# Characterizing Biofilm Interactions between *Ralstonia insidiosa* and *Chryseobacterium gleum*

Andrea Foote,[a] Kristin Schutz,[b] Zirui Zhao,[c] Pauline DiGianivittorio,[a,b] Bethany R. Korwin-Mihavics,[a] John J. LiPuma,[d] Matthew J. Wargo[b]

[a]Cellular, Molecular, and Biomedical Sciences Graduate Program, University of Vermont Larner College of Medicine, Burlington, Vermont, USA
[b]Department of Microbiology and Molecular Genetics, University of Vermont Larner College of Medicine, Burlington, Vermont, USA
[c]Department of Biology, University of Vermont, Burlington, Vermont, USA
[d]Department of Pediatrics, University of Michigan Medical School, Ann Arbor, Michigan, USA

**ABSTRACT** *Ralstonia insidiosa* and *Chryseobacterium gleum* are bacterial species commonly found in potable water systems, and these two species contribute to the robustness of biofilm formation in a model six-species community from the International Space Station (ISS) potable water system. Here, we set about characterizing the interaction between these two ISS-derived strains and examining the extent to which this interaction extends to other strains and species in these two genera. The enhanced biofilm formation between the ISS strains of *R. insidiosa* and *C. gleum* is robust to starting inoculum and temperature and occurs in some but not all tested growth media, and evidence does not support a soluble mediator or coaggregation mechanism. These findings shed light on the ISS *R. insidiosa* and *C. gleum* interaction, though such enhancement is not common between these species based on our examination of other *R. insidiosa* and *C. gleum* strains, as well as other species of *Ralstonia* and *Chryseobacterium*. Thus, while the findings presented here increase our understanding of the ISS potable water model system, not all our findings are broadly extrapolatable to strains found outside of the ISS.

**IMPORTANCE** Biofilms present in drinking water systems and terminal fixtures are important for human health, pipe corrosion, and water taste. Here, we examine the enhanced biofilm of cocultures for two very common bacteria from potable water systems: *Ralstonia insidiosa* and *Chryseobacterium gleum*. While strains originally isolated on the International Space Station show enhanced dual-species biofilm formation, terrestrial strains do not show the same interaction properties. This study contributes to our understanding of these two species in both dual-culture and monoculture biofilm formation.

**KEYWORDS** tap water microbiology, bacterial ecology, dual-species biofilms, bacterial interactions

**M**ultispecies, surface-attached biofilms are an important part of the built environment, particularly the potable water system that includes the municipal delivery system, building plumbing, terminal fixtures, appliances, and associated surfaces (1–6). Potable water system biofilms contribute to alterations in material corrosion, water properties, and health of those drinking and bathing in the water (7). While the microbial communities within these systems are diverse, there are a number of taxa that are common across wide swaths of geography and water chemistry, including the genera *Ralstonia* and *Chryseobacterium* (8). *Ralstonia insidiosa* and its close relative, *Ralstonia pickettii*, are betaproteobacteria that are common in water systems and other parts of the built environment (9, 10) and can be found infrequently as opportunistic pathogens in human infections (11–14). *Chryseobacterium gleum* (formerly, *Flavobacterium gleum*) in the phylum *Bacteroidetes*, is present in similar environments as *R. insidiosa* and can also be present in human infections (15–17), though less frequently.

Address correspondence to Matthew J. Wargo, mwargo@uvm.edu.

The authors declare no conflict of interest.

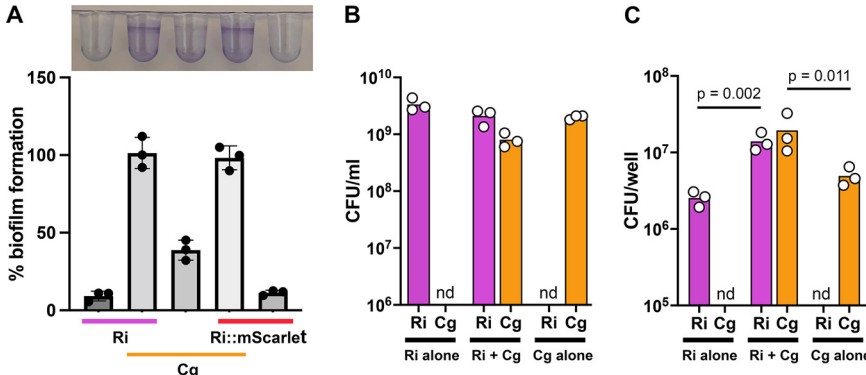

**FIG 1** Enhanced biofilm formation during *R. insidiosa* and *C. gleum* coculture. The ISS isolates of *R. insidiosa* (Ri) and *C. gleum* (Cg) were grown alone or cocultured in high-calcium MRLP medium for 24 h. (A, top) Representative crystal violet stain of these static biofilms grown in flexible 96-well dishes, with samples in the same order as the graph. (Bottom) Quantification of the crystal violet biofilm absorbance normalized to *R. insidiosa*+*C. gleum* as 100% biofilm formation. Integration of mScarlet at the *attTn7* site (Ri::*mScarlet*) does not impact *R. insidiosa*'s biofilm formation or interaction with *C. gleum*. (B) Quantification of planktonic and biofilm cells from the mono- or coculture wells. After scraping, vortexing, and resuspension, CFU were counted by serial dilution plating and subsequent counting. The $y$ axis starts at $10^6$ CFU since that was the detection limit for these dilutions. (C) Quantification of biofilm cells from the mono- or coculture wells. After standard biofilm washes, the walls were scraped and vortexed to resuspend, and the CFU were determined by serial dilution plating and subsequent counting. The $y$ axis starts at $10^5$ CFU since that was the detection limit for these dilutions. For each panel, each individual dot is the mean of three technical replicates on a separate experimental day. nd, none detected. For panel A, using ANOVA with a Sidak's posttest, significantly more biofilm was seen in cocultures compared to monocultures ($P < 0.001$) and there was also significantly more biofilm in the cocultures than the sum of the monoculture biofilms ($P < 0.005$). For panel B, using a two-way ANOVA with Tukey's posttest, the coculture was determined to not significantly impact the CFU of the individual bacteria compared to the monoculture, and the total numbers of cells per well were also not significantly altered. Data in panel C were also analyzed as for panel B, but there were significant differences in biofilm CFU for monoculture and coculture wells for both species.

*R. insidiosa* promotes biofilms in conjunction with a number of species, including *Listeria monocytogenes*, *Salmonella enterica*, and *Escherichia coli* (18–22). In most of these cases, *R. insidiosa* has been reported as a physical bridge between cells of the other species with components of the biofilm matrix predicted as driving the enhanced biofilms (21, 22). *Chryseobacterium* species are reported as strong biofilm formers in single-species cultures (17), but their enhancement of multispecies biofilms has not been reported as frequently (23).

We previously reported that strains of *R. insidiosa* isolate and *C. gleum* from the International Space Station (ISS) were capable of enhanced dual-species biofilm formation, though neither was capable of robust biofilm formation alone (24). This interaction is a critical facet for robustness of biofilm formation in this six-species model drinking water community (24). Here, we characterize the interaction between *R. insidiosa* and *C. gleum* in relation to biofilm formation, examining potential broad mechanisms, the dependence of the interaction on growth conditions, and the broader applicability of this interaction by testing other strains and species within these two genera.

## RESULTS

**Coculture of the ISS *C. gleum* and *R. insidiosa* leads to enhanced biofilm formation.** We had previously observed that the ISS isolates of *R. insidiosa* and *C. gleum* in our six-member potable water biofilm community demonstrated enhanced biofilm formation together and very little when grown alone (24). The goal of this project was to extend the characterization of this interaction. Partly to aid in this endeavor, we generated a clone of the ISS *R. insidiosa* carrying mScarlet at the *attTn7* site. Both the parent strain of *R. insidiosa* and the *R. insidiosa* with mScarlet showed enhanced biofilm formation with *C. gleum* in flexible 96-well dish biofilms (Fig. 1A). This enhanced biofilm formation is not due solely to growth stimulation of either species during coculture

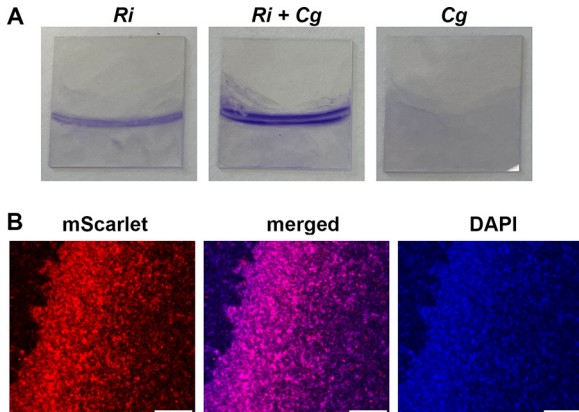

**FIG 2** Enhanced dual-species biofilm formation on glass coverslips by *C. gleum* and *R. insidiosa attTn7::mScarlet*. The ISS isolates *C. gleum* (*Cg*) and the mScarlet-expressing *R. insidiosa* (*Ri*) were grown alone or cocultured in high-calcium MRLP medium for 24 h in 12-well plates with a vertically oriented coverslip in each well. (A) Crystal violet staining of the coverslip biofilm. Note that *R. insidiosa* forms more biofilm on glass than on plastic (Fig. 1). (B) One example Z-section of the dual-species biofilm taken ~3 $\mu$m above the coverslip using confocal microscopy. The air-liquid interface runs roughly diagonally on a line rotated slightly clockwise from vertical with the air side on the left. Scale bar, 100 $\mu$m.

(Fig. 1B), though there are more adherent cells of both species during coculture compared to monoculture (Fig. 1C). There are ~13-fold more total cells in the coculture biofilm than the monoculture *R. insidiosa* biofilm and ~7-fold more total cells in the coculture biofilm than the monoculture *C. gleum* biofilm. This suggests that much of the observed biofilm enhancement is driven by increased adherent cells but does not rule out contributions of extracellular polymeric substances.

To examine the enhanced biofilm microscopically, we grew each biofilm on glass coverslips positioned roughly vertically in 12-well dishes. To visually assess biofilm formation, we stained duplicate coverslips with crystal violet. Enhanced biofilm is apparent in the cocultures, and we also note that *R. insidiosa attTn7::mScarlet* forms a stronger monoculture biofilm on glass than it does on plastic (compare Fig. 2A to Fig. 1A), which is not different than the untagged *R. insidiosa* on glass. The doublet line, most apparent in the coculture, is formed because the coverslip rests at an angle off vertical; the surface film intercepts at slightly different positions on each side of the coverslip. For fluorescence microscopy, we wiped one side thoroughly with paper cloth after fixation to remove material from the side not being viewed. Imaging of the coculture biofilm showed red-fluorescent cells that were co-stained with DAPI (*R. insidiosa*), as well as cells fluorescing solely with DAPI (4′,6′-diamidino-2-phenylindole; *C. gleum*) (Fig. 2B). Like other examples of *R. insidiosa* enhancing biofilm formation (19–22), this dual-species biofilm is intermixed.

**Impact of calcium concentration on *C. gleum* biofilm formation.** During compilation of the data and final sets of experimentation, it was noticed that while enhanced biofilm was always obvious between these two strains of bacteria, the baseline biofilm formation by *C. gleum* was very different between different experimenters and compared to our previous work (24). A deep dive through lab notebooks uncovered the likely difference as two different recipes for our MOPS-based MRLP media (see Materials and Methods) in the lab that differed only in their final calcium concentrations: 21.1 $\mu$M versus 400 $\mu$M. To test whether this calcium difference was able to explain the *C. gleum* biofilm differences between experimenters, we tested freshly made versions of both MOPS (morpholinepropanesulfonic acid) media recipes. High calcium increases *C. gleum* biofilm formation (Fig. 3). There is ~5-fold greater biofilm formation by *C. gleum* grown in high calcium versus in low calcium. The dual-species biofilms were also enhanced by high-calcium growth, but biofilm for the *R. insidiosa* monoculture remained unchanged. Because we saw enhanced biofilm at either calcium level, we present data from both calcium conditions in this paper and note the calcium level in the figure legends.

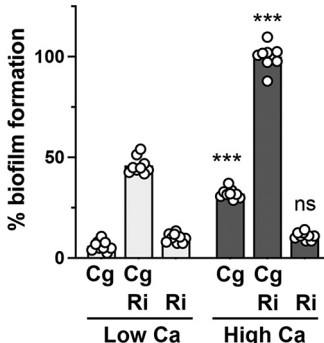

**FIG 3** Impact of calcium on biofilm formation by *C. gleum*. More biofilm is generated by *C. gleum* and the dual species biofilm when grown in high calcium (400 $\mu$M) MOPS media versus low calcium (21.1 $\mu$M) MOPS media. The calcium concentration did not impact *R. insidiosa* biofilm formation. Dots represent individual experimental replicates from three independent experiments; statistics were calculated using the means of each experiment. Two-way ANOVA with a Sidak's multiple-comparison test was used to separately compare each species composition between the two calcium conditions (***, $P < 0.001$; ns, not significant).

**The biofilm stimulation between these strains is robust to starting inoculum.** Our initial studies used equal starting optical densities (ODs) for each species, and we chose to next examine the robustness of the enhanced biofilm formation to starting inoculum. If we changed the ratio of the starting inoculum of *C. gleum* to *R. insidiosa* from 10% *C. gleum* (90% *R. insidiosa*) to 90% *C. gleum* (10% *R. insidiosa*), where 50% *C. gleum* is our previously reported 1:1 mixture, we saw that all combinations resulted in enhanced biofilm formation compared to *C. gleum* alone (100% *C. gleum*), particularly at the lower proportions of *C. gleum* (Fig. 4A). Further dilution of *C. gleum* compared to a set proportion of *R. insidiosa* showed that some biofilm enhancement of the coculture compared to that of *R. insidiosa* alone was retained down to 0.0001% *C. gleum* (roughly 15 *C. gleum* cells per well at the time of inoculation) (Fig. 4B).

**Medium and environmental impacts on the *C. gleum-R. insidiosa* interaction.** After observing that the enhanced biofilm formation in the *C. gleum-R. insidiosa* coculture was very robust to the starting inoculum (Fig. 4) and that calcium concentration could alter *C. gleum* monoculture and coculture biofilm formation (Fig. 3), we expanded the parameters and examined the impacts of medium composition, temperature, and low-shear simulated microgravity conditions.

In an initial attempt to simplify the media composition, replacement of the MOPS media in our standard MRLP with M9 (M9-RLP) or M63 (M63-RLP) was attempted, as was the RL composition (see Materials and Methods and Methods for precise descriptions). When overnight cultures were pregrown in MRLP and used to inoculate biofilm formation in various media, all tested media supported some level of enhanced biofilm between *R. insidiosa* and *C. gleum* (Fig. 5A). However, when overnights were pregrown in M9-RLP, biofilm enhancement was only seen in MRLP and RL media, while no significant biofilm enhancement was seen with M9-RLP or M63-RLP (Fig. 5B). In data not shown here, pregrowth in RL media was similar to MRLP, while pregrowth in M63-MRLP was similar to M9-RLP. One of the primary differences between MRLP/RL versus M9-RLP/M63-RLP is phosphate concentration, but we have not formally tested whether phosphate is the contributing factor in these media differences, though phosphate concentration has been well described to impact biofilm formation in many bacteria (25, 26), including in the context of drinking water systems (27).

We also tested whether enhanced biofilm formation by *C. gleum* and *R. insidiosa* was achieved at higher temperatures. Both species are opportunistic pathogens and thus capable of growth at body temperature. When grown at 37°C, similar biofilm enhancement between these species was observed as when grown at 30°C (Fig. 5C).

Finally, we focused in these studies on two strains collected from the ISS and thus previously subjected to continuous microgravity and the low-shear environment that

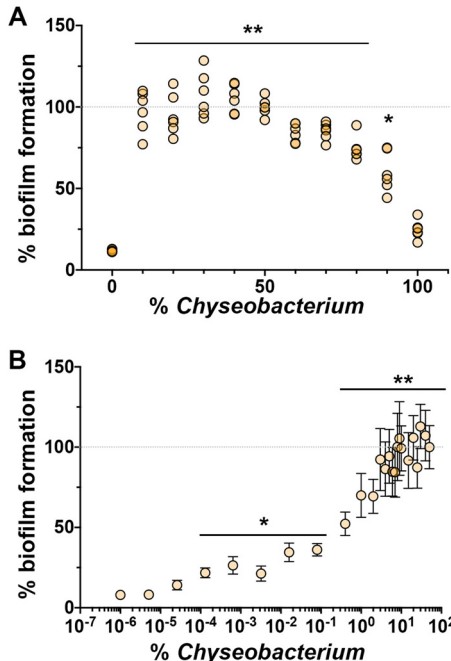

**FIG 4** Biofilm stimulation with varying initial concentrations of *C. gleum*. The ISS *C. gleum* was added as a percentage of the initial starting $OD_{600}$ noted on the *x* axis with the remainder being comprised of ISS *R. insidiosa*. All experiments conducted in high-calcium MOPS. (A) *C. gleum* stimulation of dual-species biofilm formation decreases as the *C. gleum* composition increases above 50% but shows stable stimulation as *C. gleum* composition drops below 50%. Each dot represents the measurement of a biological replicate from three separate experiments (two replicates per experiment). (B) Dilution to extinction of *C. gleum* shows biofilm stimulation above *R. insidiosa* alone (the zero *C. gleum* condition is set as the $10^{-6}$ dot), showing that biofilm stimulation can be seen with as little as 0.0001% *C. gleum*. Dots represent the mean of at least three independent experiments, and the error bars represent standard deviations. The gray dotted line marks the 100% biofilm formation that is set based on the 50/50 *R. insidiosa-C. gleum* mix, and all other biofilm formation is normalized to this 100% mark. Data were analyzed with Browne-Forsythe and Welch ANOVA with a Dunnett's posttest with 100% *C. gleum* as the comparator in panel A and 100% *R. insidiosa* as the comparator in panel B (\*, $P < 0.05$; \*\*, $P < 0.01$).

accompanies such gravity conditions. To mimic the low-shear component of microgravity, we used rotating wall vessels (RWVs) where the cells in suspension have defined vertical circular paths within a low-shear environment. While some clumping was seen for the cells in suspension in the dual-species RWV cultures, we noted that most of the biomass was present as biofilm on the gas-permeable membrane of the RWV chamber. We removed the biofilm from the membranes with silicon cell-scrapers and crystal violet stained the collected cells to assess total membrane-attached biofilms. Under these RWV conditions, enhanced biofilm formation was readily apparent compared to the individual species (Fig. 5D). Note that in the RWV system, *R. insidiosa* forms a more robust biofilm than in most other conditions reported in this study, reminiscent of its stronger biofilm on glass (Fig. 2A).

**Testing potential mechanisms of biofilm enhancement.** Most of the previous descriptions of *R. insidiosa* stimulation of biofilm in mixed cultures showed that the effect was contact dependent and not coaggregation based. Our microscopy supported comingling of the two species (Fig. 2B), but we also wanted to formally test whether a soluble factor or coaggregation could explain the enhanced biofilm formation of this specific interaction. To test whether biofilm stimulation was transferrable by conditioned media, we used cell-free supernatants from *C. gleum*, *R. insidiosa*, or cocultures mixed 1:1 with fresh media and measured the biofilm formation in the flexible 96-well format. Neither single-species nor dual-species supernatants were able to stimulate biofilm formation (Fig. 6A). This supports the direct interaction model but does not fully rule out potential for a soluble mediator, especially if the mediator is short-lived.

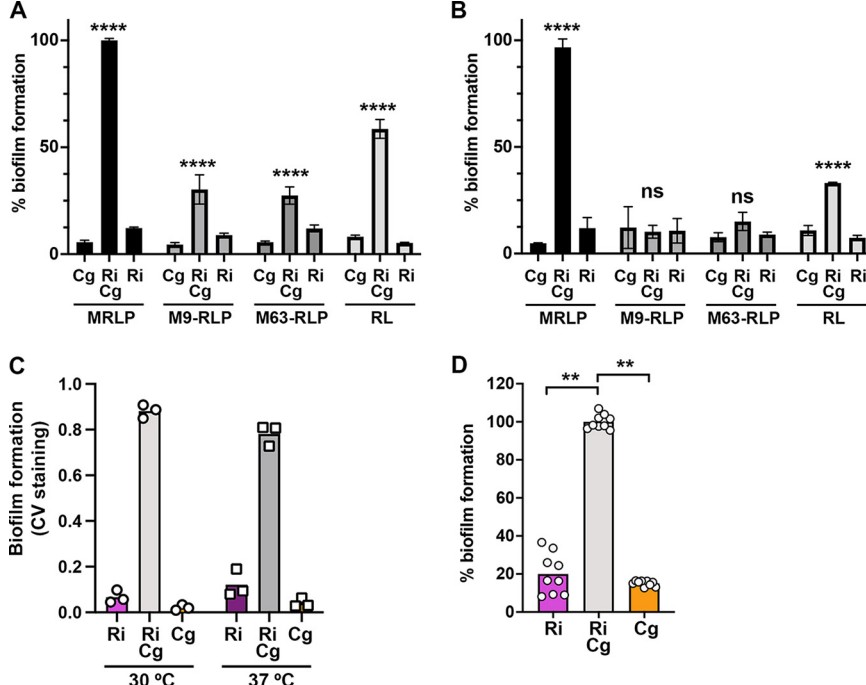

**FIG 5** Characterization of growth media, temperature, and low-shear growth on enhanced biofilm formation. (A) Biofilm formation normalized to Ri+Cg MRLP for the media listed below the *x* axis, with cells pregrown in low-calcium MRLP. (B) Normalized biofilm formation in the media listed below the *x* axis, with cells pregrown in low calcium M9-RLP. Data in panel B are normalized to Ri+Cg in MRLP from panel A. Means are from three independent experimental days; error bars represent the standard deviations. (C) Biofilm formation from the noted single or dual-species cultures in low-calcium MRLP grown at 30°C (left) or 37°C (right). Each point represents the mean of three biological replicates for each of three independent experimental days. (D) Biofilm formation on the gas-permeable membrane of a rotating wall vessel (RWV) in high-calcium MRLP. Each point represents each replicate of the RWV experiment, though statistics were calculated from each experimental day's mean (i.e., *n* = 3). Statistics for panels A to C were assessed using two-way ANOVA with a Dunnett's corrected multiple-comparison posttest comparing each single species biofilm to the dual-species biofilm within each medium or condition. Statistics for panel D were assessed using Welch's ANOVA with Dunnett's posttest comparing each single species biofilm to the dual-species biofilm.

We assessed the potential for coaggregation by mixing concentrated cultures of both species together and looking for any sedimentation caused by aggregation. At 1 h, there was no evidence of sedimentation (Fig. 6B) and no sedimentation indicative of aggregation was apparent at times up to 24 h (data not shown). Coaggregation is a specialized form of contact-dependent biofilm formation that is often typified by high-affinity interactions and, while best studied in multispecies dental biofilms (28), there are also aquatic and drinking water examples (29–31). There are many forms of contact-dependent or short distance biofilm interactions that are not driven by the same mechanism as coaggregation.

**Species and strain specificity for dual-species biofilm enhancement.** Our primary goal was to characterize the interaction between the ISS *C. gleum* and *R. insidiosa* to understand our potable water model community. However, we also wanted to understand how broadly applicable this interaction was for other species in these genera and other strains of these species. We acquired non-ISS strains of *R. insidiosa* and *R. pickettii*, an unnamed *Ralstonia* sp., *C. gleum*, *C. indologenes*, and *C. meningisepticum* (Table 1) and conducted biofilm assays in the flexible 96-well dish format. Most of the *R. insidiosa* strains could enhance ISS *C. gleum* biofilm formation, while only some of the other two *Ralstonia* species were able to do this (Fig. 7A and B). The ability of ISS *C. gleum* to respond to the *Ralstonia* presence with enhanced biofilm formation, however, was limited only to ISS *C. gleum* and very mild stimulation of *C. meningisepticum* biofilm (Fig. 7A and C). A notable difference between the ISS *C. gleum* and the other strains is

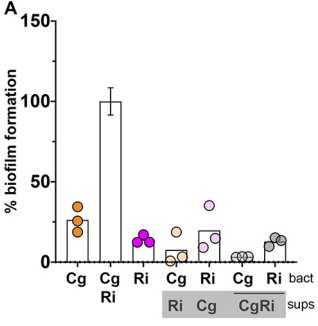

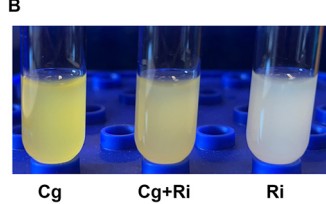

**FIG 6** The biofilm enhancing effect of either species cannot be transferred by supernatant and is not due to coaggregation. (A) The enhancement in biofilm formation seen in the dual-species growth is not conferred by cell-free supernatant from either the single or dual-species conditions using 50% spent supernatant and 50% fresh media. None of the supernatant treatments were significantly different than their single species growth after correction for multiple comparisons using a Brown-Forsythe and Welch ANOVA with a Dunnett's posttest comparing all means to each other. The source of the supernatants tested is present in the gray box under the species abbreviation. Each dot represents the mean from three biological replicates per experiment. The Cg/Ri conditions is set to 100% for each experiment to normalize between experiments, so the average standard deviation for 50/50 Cg-Ri mixes is displayed. (B) There is no coaggregation apparent between these strains. Cells grown in the high-calcium MOPS media were collected by centrifugation and resuspended at a final concentration of $OD_{600}$ = 6 and examined over time with the picture representative of all replicates at 1 h postmixing.

that the other *Chryseobacterium* spp. tested, particularly the *C. gleum* strains, form robust biofilms on their own (Fig. 7A).

## DISCUSSION

Here we examined the interaction between ISS-derived *R. insidiosa* and *C. gleum* to better understand the enhanced biofilm of the dual-species cultures (Fig. 1) as part of community biofilm formation in our six-species potable water community (24). The enhanced biofilm formation between these two ISS strains is robust to starting inoculum (Fig. 4) and temperature (Fig. 5C) does not occur in all growth media (Fig. 5B) and does not appear to be driven by a soluble mediator or coaggregation (Fig. 6). While these findings shed light on the ISS *R. insidiosa* and *C. gleum* interaction, such enhancement is not a common interaction outcome between *Ralstonia* species and terrestrial strains of *C. gleum*, as well as other *Chryseobacterium* spp. (Fig. 7). Thus, while the work presented here increases our understanding of the potable water model system, we fully acknowledge that most of the findings are not broadly extrapolatable to strains found outside of the ISS.

The enhanced biofilm formation between the ISS *R. insidiosa* and *C. gleum* is readily apparent (Fig. 1A), is not driven by changes in total cell number or proportional changes in cell number in the cultures (Fig. 1B), and is driven by increased number of adhered cells of both species (Fig. 1C). The material presented here is part of a broader characterization of the ISS potable water model community and thus has been under way for quite a number of years and with many different graduate, undergraduate, and technician contributors. It was during data examination for this report that we realized that there were two different MOPS medium recipes coexisting in the lab differing only by their calcium concentrations. This calcium difference was sufficient to describe the higher biofilm formation by *C. gleum* monoculture observed by some lab

**TABLE 1** Description of strains used in this study

| Strain | Species | Source[a] | ID[b] | Collection |
|---|---|---|---|---|
| *Ralstonia* spp. | | | | |
| MJ602 | *R. insidiosa* | ISS PW | 130770013-1 | NASA JSC |
| MJ875 | *R. insidiosa* | CF sputum | AU13589 | CFSS-UMich |
| MJ880 | *R. insidiosa* | CF sputum | AU16942 | CFSS-UMich |
| MJ885 | *R. insidiosa* | Liver biopsy | AU19393 | CFSS-UMich |
| MJ886 | *R. insidiosa* | CF sputum | AU20290 | CFSS-UMich |
| MJ890 | *R. insidiosa* | CF sputum | AU20795 | CFSS-UMich |
| MJ891 | *R. insidiosa* | CF sputum | AU21215 | CFSS-UMich |
| MJ899 | *R. insidiosa* | CF sputum | AU2944 | CFSS-UMich |
| MJ912 | *R. insidiosa* | CF sputum | AU37532 | CFSS-UMich |
| MJ913 | *R. insidiosa* | CF sputum | AU40266 | CFSS-UMich |
| MJ915 | *R. insidiosa* | CF sputum | AU42598 | CFSS-UMich |
| MJ916 | *R. insidiosa* | Blood | AU5199 | CFSS-UMich |
| MJ918 | *R. insidiosa* | CF sputum | AU9058 | CFSS-UMich |
| MJ919 | *R. insidiosa* | CF sputum | AU9775 | CFSS-UMich |
| MJ926 | *R. insidiosa* | Water | HI4151 | CFSS-UMich |
| MJ873 | *R. pickettii* | Dialysis fluid | AU10444 | CFSS-UMich |
| MJ876 | *R. pickettii* | CF sputum | AU14824 | CFSS-UMich |
| MJ877 | *R. pickettii* | Deep wound | AU15492 | CFSS-UMich |
| MJ882 | *R. pickettii* | CF sputum | AU17622 | CFSS-UMich |
| MJ884 | *R. pickettii* | CF sputum | AU17695 | CFSS-UMich |
| MJ887 | *R. pickettii* | CF sputum | AU20306 | CFSS-UMich |
| MJ892 | *R. pickettii* | Blood | AU21292 | CFSS-UMich |
| MJ895 | *R. pickettii* | CF sputum | AU27711 | CFSS-UMich |
| MJ897 | *R. pickettii* | CF swab | AU28427 | CFSS-UMich |
| MJ898 | *R. pickettii* | CF sputum | AU28504 | CFSS-UMich |
| MJ900 | *R. pickettii* | CF throat | AU30485 | CFSS-UMich |
| MJ911 | *R. pickettii* | CF sputum | AU36159 | CFSS-UMich |
| MJ914 | *R. pickettii* | CF ET aspirate | AU40593 | CFSS-UMich |
| MJ917 | *R. pickettii* | Pericardium | AU6150 | CFSS-UMich |
| MJ920 | *R. pickettii* | Soil | HI2644 | CFSS-UMich |
| MJ921 | *R. pickettii* | IV saline | HI2937 | CFSS-UMich |
| MJ922 | *R. pickettii* | Environmental | HI3561 | CFSS-UMich |
| MJ923 | *R. pickettii* | Environmental | HI3575 | CFSS-UMich |
| MJ924 | *R. pickettii* | Environmental | HI3597 | CFSS-UMich |
| MJ925 | *R. pickettii* | Environmental | HI3631 | CFSS-UMich |
| MJ874 | *Ralstonia* sp. 1 | CF sputum | AU11313 | CFSS-UMich |
| MJ879 | *Ralstonia* sp. 2 | Blood | AU15928 | CFSS-UMich |
| MJ881 | *Ralstonia* sp. 2 | ET aspirate | AU17267 | CFSS-UMich |
| MJ883 | *Ralstonia* sp. 2 | CF sputum | AU17666 | CFSS-UMich |
| | | | | |
| *Chryseobacterium* spp. | | | | |
| MJ601 | *C. gleum* | ISS PW | 113330055-2 | NASA JSC(cite) |
| MJ878 | *C. gleum* | CF sputum | AU15601 | CFSS-UMich |
| MJ893 | *C. gleum* | CF sputum | AU23904 | CFSS-UMich |
| MJ894 | *C. gleum* | CF throat | AU25398 | CFSS-UMich |
| MJ896 | *C. gleum* | CF sputum | AU28136 | CFSS-UMich |
| MJ901 | *C. gleum* | CF sputum | AU32207 | CFSS-UMich |
| MJ902 | *C. gleum* | CF nasopharynx | AU32450 | CFSS-UMich |
| MJ903 | *C. gleum* | CF throat | AU33652 | CFSS-UMich |
| MJ904 | *C. gleum* | CF sputum | AU33758 | CFSS-UMich |
| MJ905 | *C. gleum* | CF nasopharynx | AU34337 | CFSS-UMich |
| MJ907 | *C. gleum* | CF sputum | AU34937 | CFSS-UMich |
| MJ908 | *C. gleum* | CF | AU35407 | CFSS-UMich |
| MJ910 | *C. gleum* | CF sputum | AU36080 | CFSS-UMich |
| MJ906 | *C. indologenes* | CF throat | AU34532 | CFSS-UMich |
| MJ909 | *C. indologenes* | CF sputum | AU35927 | CFSS-UMich |
| MJ888 | *C. meningisepticum* | CF sputum | AU20341 | CFSS-UMich |
| MJ889 | *C. meningisepticum* | CF sputum | AU20342 | CFSS-UMich |

[a]PW, potable water; CF, sputum from a person with cystic fibrosis; ET, endotracheal; IV, intravenous.
[b]The database ID at the source repository. NASA JSC(cite), National Aeronautics and Space Administration Johnson Space Center; CFSS-Umich, Cystic Fibrosis Stock Center- University of Michigan.

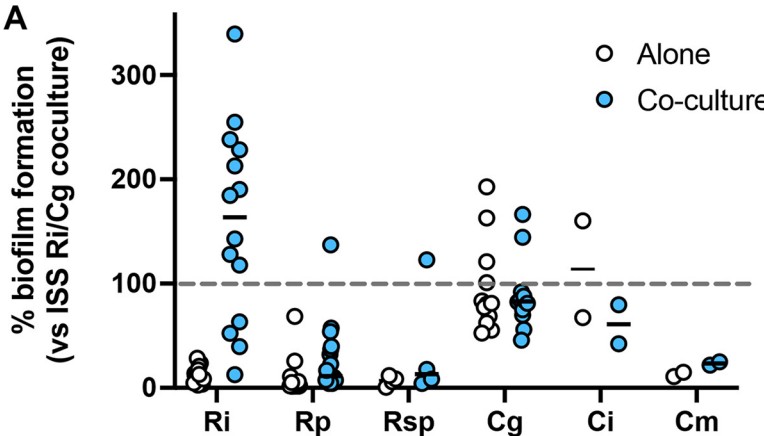

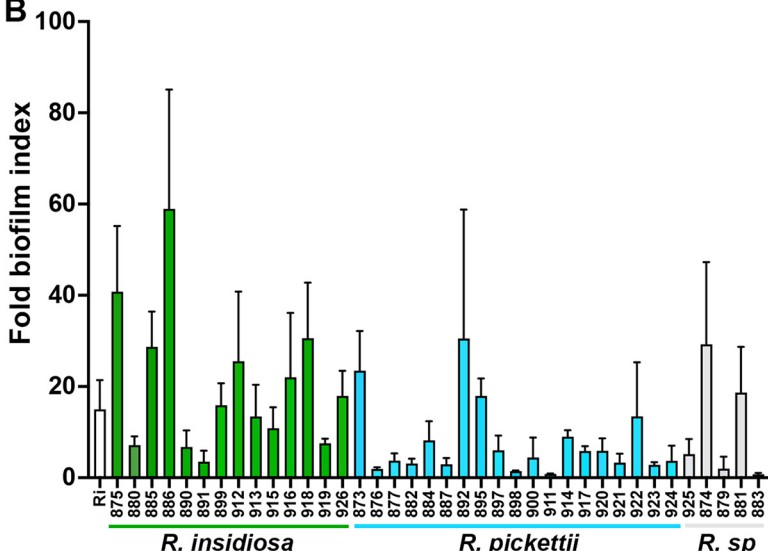

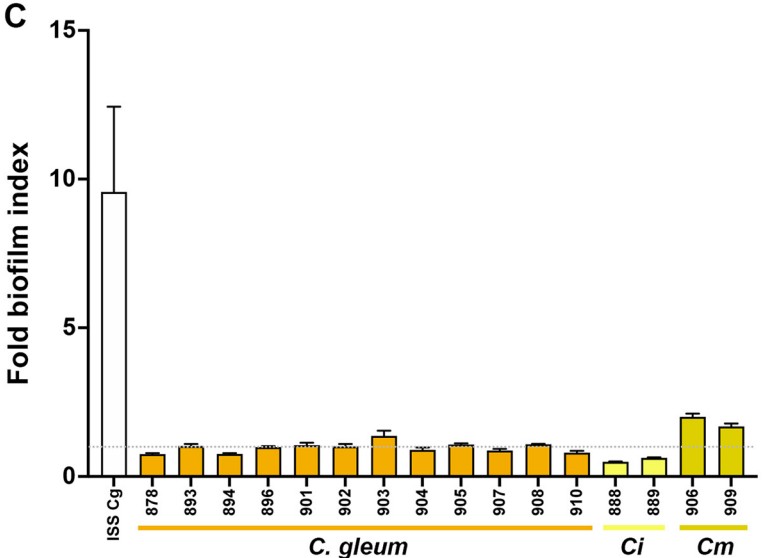

**FIG 7** Strain and species specificity for dual-species biofilm enhancement. (A) Biofilm quantification normalized by setting the intraday ISS *R. insidiosa* and ISS *C. gleum* dual-species biofilm as 100% biofilm formation. White circles are data for each species alone, and the blue circles are in the presence of either the ISS *C. gleum* (for *Ralstonia* species) or the ISS *R. insidiosa* (for the *Chryseobacterium* species). Two-way

members but not others (Fig. 3). The general biofilm enhancement between the two ISS bacteria is still readily apparent at either calcium concentration, and the calcium concentration of the resultant MRLP media in each experiment is noted in each figure legend.

The enhancement of biofilm formation between the ISS *R. insidiosa* and *C. gleum* is very robust to starting inoculation (Fig. 4). While we did not assess resulting CFU for each species in these broader inoculation tests, we have no *a priori* reason to suspect that there are different mechanisms of enhancement at different starting inocula, though we cannot rule this out. The enhanced biofilm formation was seen at both 30 and 37°C (Fig. 5C) and occurred at room temperature (22 to 24°C) in the RWV experiments (Fig. 5D). The RWVs were used to ensure that the biofilm enhancement we observed in open, static biofilms was retained in a system mimicking the low-shear environment found during growth in microgravity, as applicable to the ISS. While enhanced biofilm formation could happen with surprisingly low inoculum of *C. gleum*, no enhancement of biofilm could be seen in either direction by single culture or coculture supernatants (Fig. 6A), suggesting that there is either no soluble mediator or that any such mediator is very labile. Coaggregation, while a common mechanism in biofilms, particularly in the oral cavity, does not drive the enhanced biofilm formation in this interaction (Fig. 6B).

When the ISS strains were grown in MRLP in the overnight cultures used for inoculation, there was observable dual-species biofilm enhancement in all tested media (Fig. 5A), but when cells were instead grown in overnights of M9-RLP (Fig. 5B) or M63-RLP (not shown), there was no biofilm enhancement in the subsequent M9-RLP or M63-RLP conditions, though enhancement was retained in MRLP and RL. We suspect that this is due to the relatively high phosphate concentration in the M9 and M63 media that could be suppressing biofilm enhancement. However, mechanistic dissection of the media component contribution to biofilm enhancement was halted once we determined that this interaction was not commonly shared between strains of these species (Fig. 7).

The ISS *R. insidiosa* behaves similarly to most of the other *R. insidiosa* strains in terms of their weak monoculture biofilm formation and ability to enhance biofilm when grown with ISS *C. gleum* (Fig. 7). Most of the *R. pickettii* strains do not show strong biofilm enhancement with *C. gleum*, nor does a *Ralstonia pickettii*-like species that has not yet been named. These similarities among the *R. insidiosa* strains suggest the ISS *R. insidiosa* and our mScarlet labeled derivative are potentially good representatives of the species. It is at least minimally genetically tractable, since we were able to generate an *attTn7::mScarlet* integrant that may be useful for studying *R. insidiosa* interaction studies with other bacteria.

The non-ISS *C. gleum* strains form strong biofilms, as shown in Fig. 7 and from literature reports (17), as do the two tested *C. indologenes* strains, while the two tested *C meningisepticum* strains do not. Thus, the ISS *C. gleum* appears to be nonrepresentative of other *C. gleum* strains. While all our *Chryseobacterium* isolates tested were from clinical sources, others in the literature were isolated from environmental sources and also show strong biofilm formation in monoculture. Thus, this ISS *C. gleum* appears only useful to describe its specific interactions among the ISS potable water bacteria. Whether the specific biofilm phenotype of the ISS *C. gleum* is a response to the condi-

**FIG 7** Legend (Continued)

repeated measures ANOVA with Sidak's multiple-comparison test comparing intra-species mono- and dual-species biofilms supports that, at the species level, only *R. insidiosa* shows significant biofilm enhancement under dual-species conditions ($P < 0.001$). (B) The same data contributing to panel A converted to the fold change of dual-species biofilms divided by biofilm formation by that strain alone. Green bars represent the *R. insidiosa* strains, blue bars represent *R. pickettii* strains, and gray bars represent an unnamed *Ralstonia* species. (C) The same data contributing to panel A represented as a fold change of dual species biofilm divided by biofilm formation by that strain alone. The white bar is the ISS *C. gleum*, orange bars are the non-ISS *C. gleum* strains, yellow bars are *C. indologenes* strains, and mustard bars are *C. meningisepticum* strains.

tions in the ISS potable water system or to prolonged coculture with the other ISS microbes in a small closed system is unknown. It is tempting to speculate that loss of monoculture biofilm formation would be promoted by the low-flow, low-shear microgravity environment and the interaction with a bottlenecked microbial population and no immigration of new drinking water bacteria.

In conclusion, we have presented partial dissection of an important interaction in a model potable water community. Despite the observation that this strain-specific interaction is unlikely to be broadly applicable beyond this model, we have provided additional information on basal biofilm formation among diverse *C. gleum* and *R. insidiosa* strains and generated an mScarlet-expressing *R. insidiosa* that may be useful to others in the field or studying *R. insidiosa* pathogenesis.

## MATERIALS AND METHODS

**Bacterial strains and maintenance conditions.** *R. insidiosa* 130770013-1 was isolated from the ISS Potable Water Delivery System, and *C. gleum* 113330055-2 was isolated from the Russian SVO-ZV module on the ISS; both were identified by the Microbiology Laboratory at the NASA Johnson Space Center (Houston, TX) (1). The strain numbers listed after each species are for the strain database at the Johnson Space Center, and strains may be requested directly using these designations. Terrestrial strains of *R. insidiosa*, *R. pickettii*, *C. gleum*, *Chryseobacterium indologenes*, *Chryseobacterium meningosepticum*, and *Ralstonia* strains that cannot be identified as belonging to one of the currently named species in this genus were acquired from the Cystic Fibrosis Foundation *Burkholderia cepacia* Research Laboratory and Repository at the University of Michigan. Details on all strains presented in Table 1. All bacteria were stored in 20% glycerol stocks at $-80°C$ and were recovered on R2A plates at 30°C.

For all experiments below, the R2A plates were used as a source of inoculation into MRLP media (24) (two-third modified MOPS media [32], one-third R2B, with 2% Luria-Bertani medium [LB] and 10 mM sodium pyruvate; in composition, this medium comprises 26.4 mM MOPS, 2.64 mM tricine, 6.28 mM $NH_4Cl$, 34.7 mM NaCl, 0.35 mM $MgCl_2$, 0.19 mM $K_2SO_4$, 1.57 mM $K_2HPO_4$, 10.91 mM sodium pyruvate, 0.94 mM glucose, 32 $\mu$M $MgSO_4$, 21.1 $\mu$M $CaCl_2$, 6 $\mu$M $FeSO_4$, 5.28 $\mu$M $FeCl_2$, 0.33 g $L^{-1}$ peptone, 0.27 g $L^{-1}$ yeast extract, 0.20 g $L^{-1}$ tryptone, 0.17 g $L^{-1}$ soluble starch, and $0.66\times$ MOPS medium micronutrients), and cells were grown overnight in 3 mL of culture at 30°C with orbital shaking of angled, loosely capped 18 $\times$ 150-mm glass tubes. Some experiments were also conducted in a high-calcium version of MRLP where the final $CaCl_2$ concentration was 400 $\mu$M. The higher calcium concentration increases monoculture biofilm formation for *C. gleum*, as described in Results, but coculture enhancement is still apparent. The calcium concentration is noted in each figure legend.

Other medium formulations were also tested to assess the range of conditions within which coculture biofilm enhancement could be seen. These included versions of MRLP where the a two-third volume of minimal media was switched from MOPS to M9 or M63 to generate M9-RLP and M63-RLP, respectively. In addition, RL media was used which was one-third R2B with 2% LB with the remainder of the volume made up from distilled-deionized water.

**Biofilm initiation, quantification, and cell numbers.** For 96-well format static biofilms, overnight MRLP-grown cultures were collected by centrifugation, washed once with the destination media, and resuspended in the destination media. The OD of each species was determined and normalized to generate an initial OD at 600 nm ($OD_{600}$) of 0.05 for each species member in MRLP. After pipetting 150 $\mu$L of the species mixtures into flexible 96-well plates, the plates were placed into humidified chambers and incubated under static conditions (no shaking) at 30°C for 24 h. To measure biofilm formation, the crystal violet staining protocol was used as described by O'Toole and colleagues (33). Biofilms were quantified by dissolving the crystal violet with 10% acetic acid and measuring the absorbance at 550 nm.

For rotating wall vessel (RWV) biofilms, cells were grown overnight as for the 96-well biofilms except without washing the cells and inoculated into 10-mL RWVs (Synthecon, Inc.; ethanol sterilized and dried) at a starting inoculum of 0.008 $OD_{600}$ per species. RWVs were rotated in the vertical orientation at room temperature overnight. In conducting the RWV experiments, we observed that most of the biomass, particularly in the dual-species culture, was present as a biofilm on the air-permeable membrane of the vessel. To measure these biofilms, the membrane was rinsed with sterile media and then scraped with a silicon cell scraper into a petri dish containing additional sterile media. The contents of the petri dish were then collected by centrifugation, and pellets were stained with crystal violet as described above. After staining, pellets were washed twice before dissolution of the crystal violet in 10% acetic acid. Due to the large quantity of biofilm material, the crystal violet was quantified by 2-fold serial dilutions, which were then corrected to total absorbance by multiplication with the respective serial dilution factor.

For biofilms on glass coverslips, 22-mm square coverslips were sterilized in ethanol and dried prior to insertion into 12-well plates. The coverslips fit snugly in the wells but rest slightly off vertical. Each well contained 2.2 mL of high-calcium MRLP, which was enough to reach about halfway up the coverslip. After inoculation as described for the 96-well biofilms, the 12-well dishes were incubated statically at 30°C in a humidified chamber for 20 to 24 h.

To determine total cell number in the static 96-well plates (planktonic plus biofilm), cells were gathered by resuspending the contents of each well via vigorous pipetting and scraping the well walls with a pipette tip. As measured by crystal violet staining, <1% of the biomass remained after this treatment. The scraped well contents were vortexed to resuspend and serially diluted in R2B before plating onto

R2A. For wells with both species, CFU counts for each species were determined by the obvious colony color difference (orange-yellow for *C. gleum*, and white-beige for *R. insidiosa*).

**Statistical analysis and data visualization.** All data visualization and statistical analyses were conducted in GraphPad Prism v9 using one-way or two-way analysis of variance (ANOVA) with Tukey, Sidak, Bonferroni, or Dunnett's posttesting, as described in individual figures. *P* values of <0.05 were considered statistically significant. The data are presented, when possible, with bars representing the means and either biological replicates or all experimental replicates shown as individual data points overlain. Even when all experimental replicates are depicted, statistical analysis was conducted using the means of each biological replicate only. Where individual data points are not shown for clarity and figure size considerations, error bars represent standard deviations.

**Generation of mScarlet-expressing *R. insidiosa*.** We used *attTn7*-based chromosomal integration to generate stable fluorescent *R. insidiosa* using the general method for Gram-negative bacteria (34). Briefly, we mixed *R. insidiosa* with *Escherichia coli* S17-1 carrying pMRE-Tn*7*-145 (34) and allowed conjugation to occur on an LB plate with 0.1% L-arabinose overnight. The spot was scraped and resuspended, and *R. insidiosa* carrying the *attTn7* insertion was selected by plating on MOPS agar with 25 mM pyruvate, 10 mM glucose, 25 $\mu$g/mL gentamicin, and 10 $\mu$g/mL chloramphenicol. Colonies were picked and screened for presence of the *R. insidiosa*-specific PCR product (11), the integration cassette via PCR for the gentamicin resistance gene (GmR-check-F [5′-TCTTCCCGTATGCCCAACTT-3′] and GmR-check-R [5′-ACCTACTCCCAACATCAGCC-3′]), and fluorescence concordant with mScarlet (using 565-nm excitation and 600-nm emission with a BioTek H1 multimode plate reader).

**Microscopy.** Fluorescence microscopy was carried out on a Nikon A1R-ER confocal laser microscope housed in the UVM Microscopy Imaging Center using the RFP laser and emission filter for mScarlet which, while not optimized for mScarlet, still allowed visualization of the mScarlet signal. Images were captured and analyzed by using Nikon NIS-Elements software.

## ACKNOWLEDGMENTS

We acknowledge the many undergraduate researchers and graduate rotation students who examined various aspects of the *R. insidiosa-C. gleum* interaction that are not reported here but still contributed holistically to our understanding of these two bacteria. These individuals include Matthew Bompastore, Sierra Bruno, Brynn Cairns, Matthew Kinahan, Alexis Nadeau, Hannah Schulman, Alex Thompson, Sophie Unger, and Trevor Wolf.

We declare that there are no conflicts of interest.

This study was supported by NASA cooperative agreements 20-EPSCoR2020-0079 and 16-EPSCoR2016-0019 to M.J.W. Graduate student rotation support for A.F. and B.R.K.-M. was provided by the University of Vermont Cellular, Molecular, and Biomedical Sciences (CMB) Program. P.D. was supported by the Vermont Lung Center Multidisciplinary Training in Lung Biology T32 HL076122. Z.Z. was supported by a summer undergraduate research fellowship from the University of Vermont. Confocal microscopy was performed in the Microscopy Imaging Center at the University of Vermont on a Nikon A1R-ER point scanning confocal, supported by NIH award 1S10OD025030-01 from the Office of Research Infrastructure Programs. The funders had no role in study design, data collection and interpretation, or the decision to submit the work for publication.

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
