## [Reviewer comments · Microbiology Spectrum]

Microbiology Spectrum

Characterizing biofilm interactions between *Ralstonia insidiosa* and *Chryseobacterium gleum*

Andrea Foote, Kristin Schutz, Zirui Zhao, Pauline DiGianivittorio, Bethany Korwin-Mihavics, John LiPuma, and Matthew Wargo

Corresponding Author(s): Matthew Wargo, University of Vermont Larner College of Medicine

Review Timeline:

Submission Date:	October 10, 2022
Editorial Decision:	November 9, 2022
Revision Received:	January 4, 2023
Accepted:	January 11, 2023

Editor: Olaya Rendueles

Reviewer(s): The reviewers have opted to remain anonymous.

Transaction Report:

DOI: <https://doi.org/10.1128/spectrum.04105-22>

November 9, 2022

Prof. Matthew J Wargo
University of Vermont Larner College of Medicine
Microbiology and Molecular Genetics
95 Carrigan Drive
304 Stafford Hall
Burlington, VT 05405

Re: Spectrum04105-22 (Characterizing biofilm interactions between *Ralstonia insidiosa* and *Chryseobacterium gleum*)

Dear Prof. Matthew J Wargo:

Thank you for submitting your manuscript to Microbiology Spectrum. Two reviewers have read your manuscript and found it interesting, easy to read and follow. Their comments are overall very positive but there are some minor modifications to attend to prior to publication.

Link Not Available

Sincerely,

Olaya Rendueles Garcia

Journals Department
Reviewer comments:

Reviewer #1 (Comments for the Author):

In this report the investigators explore *Ralstonia-Chryseobacterium* biofilm interactions. These strains come from the potable water system on the ISS and are commonly isolated. They provide a basic characterization of the interactions and assess how other strains behave in this interaction. They conclude neither secreted factors nor a co-aggregation mechanism are not required for the interaction, and that these interactions occur in many (but not all) growth conditions. The interactions appear to be

specific to ISS-adapted strains.

This is a solid piece of work, with clear, well-controlled studies. The data are clearly presented with the appropriate statistics. This work will set the stage for a system that will allow the molecular dissection of these interactions. I have a few minor comments below.

Specific comments

1. Figure 1B, I think it would be best to label the mono-culture bars at BD for below detection. The dot makes it seem like there is a detectable CFU.
2. Line 201. Might be clearer to say "(compare Figure 2A to Figure 1A)" here.
3. Line 222-223. Change "for *R. insidiosus* remained unchanged" to "for *R. insidiosus* monoculture remained unchanged".
4. Line 255. Might be worth mentioning that Pi is a known signal modulating biofilm formation. Either here or in the discussion.
5. Line 304. Replace "big" with "notable"
6. Line 374-6. "Thus, our ISS *C. gleum* appears only useful to describe its specific interactions amongst the ISS potable water bacteria." Could this suggest adaptation and developing a new trait that promotes survival in these water systems? This could be quite interesting.
7. Figure 7 does not have the "A,B,C" labels. For panels B and C I think it is better to just write out "biofilm index". You have the space.

Reviewer #2 (Comments for the Author):

In this manuscript, Foote et al characterize the parameters that lead to increased biofilm formation by two strains from a six-species biofilm community originally isolated from the International Space Station: *Ralstonia insidiosus* and *Chryseobacterium gleum*. The experiments are thorough in comparing the variables that impact this increased biofilm formation (temperature, strain diversity, media, etc.), but the authors could further clarify the nature of the increased biofilm formation.

Major comments

1. The major unclear part of this manuscript is why the biofilm formation is enhanced in the 2-species community. The total cell numbers (planktonic + adhered) stay the same, as shown in Fig 1B. But is the increased crystal violet staining due to an increased percentage of adhered cells? Or due to an increase in EPS production that increases staining? It would be helpful to quantify the percentage of bacteria in the adhered and planktonic fractions of the cultures in mono- and co-culture. If the increased cell numbers in the biofilm do not explain the increased crystal violet staining, are there more specific stains that could be used (e.g., protein- or DNA-specific extracellular stains)? It also would be useful to understand how this varies between the high and low calcium conditions.
2. Related to the previous comment, the nature of the interaction between *R. insidiosus* and *C. gleum* is unclear. Could the authors provide more insight into why these two bacteria were chosen from the previous study? And did the previous work lead to a prediction or hypothesis about how the microbes might be interacting here? Fig 6 is useful in terms of narrowing the possible mechanisms of interactions, but more insight could be provided. For instance, is one of the microbes the primary component of the biofilm?
3. Fig 2B - Is there a way to know if the DAPI staining is also staining extracellular DNA? This is not clear, and if there is extracellular DNA, then it is not possible to know if the DNA or *C. gleum* is co-localized with the *R. insidiosus*.
4. The authors' main conclusion from Fig 7A and 7B is that the increased biofilm formation is seen across different *R. insidiosus* strains. However, they contradict this finding at L316 by saying that the enhancement is not a common interaction outcome. In addition, the variation across *R. insidiosus* strains is notable in Fig 7A and 7B, and potentially could provide insight into the mechanism of the biofilm enhancement in the 2-species community. Is there anything known about the differences between these *R. insidiosus* strains that might explain the variation in co-culture biofilm formation? And if genomes are available for these strains, is there any sign of a phylogenomic pattern to the biofilm enhancement?

Minor comments

1. L103 - Please give the recipe or citation for the MOPS medium micronutrients.
2. Were the cultures washed before inoculating the biofilms? In Figure 5, an alternative interpretation to the one at L355 is that potentially there is something in MRLP or RL that is important for biofilm formation, as you only see the increased biofilms when either the overnights or the biofilm cultures were grown in one of these mediums. Also, potentially a role of pH?
3. L278 - It would be helpful to clarify the difference between contact dependent and co-aggregation in biofilms

Staff Comments:

Preparing Revision Guidelines

Please return the manuscript within 60 days; if you cannot complete the modification within this time period, please contact me. If you do not wish to modify the manuscript and prefer to submit it to another journal, please notify me of your decision immediately so that the manuscript may be formally withdrawn from consideration by Microbiology Spectrum.

We thank the reviewers for their supportive reviews and assistance in improving this study. Here are the point-by-point responses all below the comment using slightly larger green text on a line beginning with a > symbol.

Reviewer #1:

This is a solid piece of work, with clear, well-controlled studies. The data are clearly presented with the appropriate statistics. This work will set the stage for a system that will allow the molecular dissection of these interactions. I have a few minor comments below.

> We appreciate the reviewer's assessment of the study and the suggestions for improvement of the manuscript.

Specific comments

1. Figure 1B, I think it would be best to label the mono-culture bars at BD for below detection. The dot makes it seem like there is a detectable CFU.

>That is a good point. This has been fixed in the new Fig1. Fig1 also has new data requested by reviewer2 and the legend has been updated to include the new abbreviation and the new panel details.

2. Line 201. Might be clearer to say "(compare Figure 2A to Figure 1A)" here.

>Excellent suggestion. Done.

3. Line 222-223. Change "for R. insidiosa remained unchanged" to "for R. insidiosa monoculture remained unchanged".

>Much clearer. This has now been changed.

4. Line 255. Might be worth mentioning that Pi is a known signal modulating biofilm formation. Either here or in the discussion.

>Good suggestion. We have mentioned this in the text and added appropriate citations.

5. Line 304. Replace "big" with "notable"

>Done

6. Line 374-6. "Thus, our ISS C. gleum appears only useful to describe its specific interactions amongst the ISS potable water bacteria." Could this suggest adaptation and developing a new trait that promotes survive in these water systems? This could be quite interesting.

>This would indeed be very interesting. We have added some language alluding to that possibility.

7. Figure 7 does not have the "A,B,C" labels. For panels B and C I think it is better to just write out "biofilm index". You have the space.

>Agreed. We have made these changes.

Reviewer #2 (Comments for the Author):

In this manuscript, Foote et al characterize the parameters that lead to increased biofilm formation by two

strains from a six-species biofilm community originally isolated from the International Space Station: *Ralstonia insidiosa* and *Chryseobacterium gleum*. The experiments are thorough in comparing the variables that impact this increased biofilm formation (temperature, strain diversity, media, etc.), but the authors could further clarify the nature of the increased biofilm formation.

>We thank the reviewer for the evaluation and have attempted to clarify as much as we are able with the responses to the comments below and the text and data added to the manuscript.

Major comments

1. The major unclear part of this manuscript is why the biofilm formation is enhanced in the 2-species community. The total cell numbers (planktonic + adhered) stay the same, as shown in Fig 1B. But is the increased crystal violet staining due to an increased percentage of adhered cells? Or due to an increase in EPS production that increases staining? It would be helpful to quantify the percentage of bacteria in the adhered and planktonic fractions of the cultures in mono- and co-culture. If the increased cell numbers in the biofilm do not explain the increased crystal violet staining, are there more specific stains that could be used (e.g., protein- or DNA-specific extracellular stains)? It also would be useful to understand how this varies between the high and low calcium conditions.

>We agree that a mechanistic understanding of the interaction driving this enhanced biofilm would be optimal and that was our original trajectory before realizing that the ISS *Chryseobacterium* was not representative of terrestrial strains in terms of biofilm interactions. We do agree with you, however, that some more information was warranted beyond what was initially presented. The biofilm is driven by more adhered cells, which we have now presented in a new panel as Fig1C with associated description in the Results. There are roughly equal CFU of each species under co-culture and the total adhered CFU in co-culture is 13x higher than Ri alone and 7x higher than Cg alone. If we look at the CV stained coverslips (Fig2A) under the scope, all apparent CV is associated with bacterial cells – which doesn't eliminate the possibility of EPS, but means that if there is EPS it is tightly cell associated. A quick check of the co-culture biofilms with the extracellular DNA dye YOYO-1 revealed no abundant eDNA.

2. Related to the previous comment, the nature of the interaction between *R. insidiosa* and *C. gleum* is unclear. Could the authors provide more insight into why these two bacteria were chosen from the previous study? And did the previous work lead to a prediction or hypothesis about how the microbes might be interacting here? Fig 6 is useful in terms of narrowing the possible mechanisms of interactions, but more insight could be provided. For instance, is one of the microbes the primary component of the biofilm?

>These two bacteria were chosen from the initial study because they showed strong co-culture biofilm enhancement and the interaction was one of the critical ones for the robustness of the community biofilm formation. This is part of the explanation in the last paragraph of the introduction. This was one of two strong co-culture interactions in that multispecies community.

3. Fig 2B - Is there a way to know if the DAPI staining is also staining extracellular DNA? This is not clear, and if there is extracellular DNA, then it is not possible to know if the DNA or *C. gleum* is co-localized with the *R. insidiosa*.

>As mentioned in the response to comment #1, YOYO-1 staining shows very little eDNA. This is not uncommon in biofilms formed in low nutrient conditions, as the eDNA seems to either not be produced as much or is scavenged quickly. The increased adherent cell number (new Fig1C) appears to explain most of the enhanced CV staining.

4. The authors' main conclusion from Fig 7A and 7B is that the increased biofilm formation is seen across

different *R. insidiosus* strains. However, they contradict this finding at L316 by saying that the enhancement is not a common interaction outcome. In addition, the variation across *R. insidiosus* strains is notable in Fig 7A and 7B, and potentially could provide insight into the mechanism of the biofilm enhancement in the 2-species community. Is there anything known about the differences between these *R. insidiosus* strains that might explain the variation in co-culture biofilm formation? And if genomes are available for these strains, is there any sign of a phylogenomic pattern to the biofilm enhancement?

>We were not sufficiently clear in that sentence and have tried to make it more clear. While most *Ri* and some *Rp* strains could enhance biofilm co-cultured with ISS *Cg*, no *Ri* strains could enhance biofilm formation with terrestrial *Cg* strains. So, while the ability or extent to which *Ri* strains enhance biofilm in conjunction with ISS *Cg* is interesting, the lack of correspondence with any terrestrial *Cg* greatly decreased our interest in further examining this system and our ability to get such work funded. The genomes are available for about ¼ of these strains, but we have not done any comparative examinations.

Minor comments

1. L103 - Please give the recipe or citation for the MOPS medium micronutrients.

>The MOPS recipe, including the micronutrients, is cited in the parenthetical list that includes the micronutrients.

2. Were the cultures washed before inoculating the biofilms? In Figure 5, an alternative interpretation to the one at L355 is that potentially there is something in MRLP or RL that is important for biofilm formation, as you only see the increased biofilms when either the overnights or the biofilm cultures were grown in one of these mediums. Also, potentially a role of pH?

>Thank you for catching this lack of detail in our methods and this is a very important point. We have now included this detail in the methods section. Yes, for all of the non-RWV data in this manuscript the cells were washed with the destination media prior to inoculating the biofilm. For the RWV, we did not wash first, as the starting inoculum was 6x less concentrated than in the 96-well or 12-well formats and, in data not shown, washing did not make a difference in resultant biofilms in any format.

3. L278 - It would be helpful to clarify the difference between contact dependent and co-aggregation in biofilms

>That is a good point and we have now included a brief explanation of this difference.

January 11, 2023

Prof. Matthew J Wargo
University of Vermont Larner College of Medicine
Microbiology and Molecular Genetics
95 Carrigan Drive
304 Stafford Hall
Burlington, VT 05405

Re: Spectrum04105-22R1 (Characterizing biofilm interactions between *Ralstonia insidiosa* and *Chryseobacterium gleum*)

Dear Prof. Matthew J Wargo:

I have read your response to the reviewers concerns and i believe you have addressed all issues. I am happy to inform you that your manuscript has been accepted, and I am forwarding it to the ASM Journals Department for publication. You will be notified when your proofs are ready to be viewed.

Sincerely,

Olaya Rendueles
Editor, Microbiology Spectrum
